# Optimizing Interfaces of Construction Processes by Digitalization Using the Example of Hospital Construction in Germany

Sabine Hartmann [1,*], Dirk Gossmann [2], Suzan Kalmuk [2] and Katharina Klemt-Albert [1]

1. Institute for Construction Management, Digital Engineering and Robotics in Construction, RWTH Aachen University, 52070 Aachen, Germany
2. Faculty of Civil Engineering, RWTH Aachen University, 52062 Aachen, Germany
* Correspondence: hartmann@icom.rwth-aachen.de; Tel.: +49-2418025145

**Abstract:** In hospital construction, additional challenges must be considered, such as an increased number of stakeholders and building trades, such as medical and laboratory technology. Due to the increasing requirements and challenges, associated construction processes are becoming more intricate. Especially for complex building types, the effects of this development are clearly noticeable and cause considerable disruptions to the construction process. A main difficulty constitutes the missing definition of the interfaces of building trades and participants. In the present study, interfaces in hospital construction were identified and analyzed by guided interviews with experts from the health sector. The qualitative content analysis, according to Mayring, was used for the evaluation to derive appropriate solution approaches. This paper presents the interfaces using the example of hospital construction in Germany and general approaches of optimization. Hereby, the digital method Building Information Modeling (BIM) plays a decisive role in the optimization of interfaces, especially in complex buildings. Furthermore, a task and building trade control matrix is required to better coordinate the interfaces. The identified approach intends to alleviate potential disputes and misunderstandings among stakeholders, as well as to improve time and financial predictability, which are particularly valuable during inflationary periods.

**Keywords:** digitalization; BIM; hospital construction; interfaces; optimization; critical infrastructure; building information modeling; healthcare management; project management

## 1. Introduction

Over the past 20 years, productivity in the global economy has increased by about 2.8% annually [1]. The manufacturing industry was able to achieve an increase of 3.6%, while the construction industry could increase productivity by only 1%. In 2020, global construction recorded a volume of over $12 trillion US dollars (USD) and is expected to register a Compound Annual Growth Rate (CAGR) of about 7% until 2030 [2] (pp. 1–3). Despite positive forecasts and immense technical progress, the construction industry faces major deficits in the coordination of construction measures, resulting in longer construction times and higher costs. In order to remedy these deficits and optimize construction processes, digitalization in the construction industry is inevitable, but it poses major challenges. Nonetheless, digitalization offers great opportunities for future developments in this sector.

### 1.1. Issue

Due to the uniqueness of each building, the requirements for the realization of construction projects are highly complex nowadays. Therefore, construction projects require process optimization of the workflows already in the planning phase. This is necessary due to the scope and desired speed of realization of projects, as well as the number of stakeholders involved. Additional pressure on costs and deadlines, as well as unforeseen changes in conditions, increase the need for information and communication in construction projects. Otherwise, the quality of design and execution will be compromised [3] (p. 173).

A significant problem in construction projects is further planning during construction. Planning is often overtaken by reality, and coordination between the stakeholders is often difficult to reconcile with the construction process. Coordinating the provision of construction services is an additional and special challenge in large-scale projects [4]. Furthermore, there are supply bottlenecks due to impending resource constraints or logistical difficulties caused by the pandemic. In the following, three worst-case scenarios from projects in Germany are described, which, due to significant design errors, have led to a drastic increase in costs and a considerable delay in project completion.

A large number of design and construction errors were discovered at the new Berlin airport, which opened with a delay of nine years. First, construction work had already been commissioned, although the necessary building licence for the construction had not been issued by the local authorities. Secondly, the planning phase had not been completed before the start of construction, so modifications had to be made during the construction phase. The subsequent changes caused high costs that had not been included in the calculations. Thirdly, the construction was characterized by considerable deficiencies in execution. A particular difficulty was the fire protection, which was affected by the technical problems of the fire alarm systems and incorrect installation of cabling. Some of the construction deficiencies could be traced back to inadequate or even missing construction supervision, which delayed consultations and impeded proper interface coordination [5]. In total, the planning and construction costs increased from an estimated 1.9 billion euros (EUR) to 5.9 billion euros (EUR) in 2020, when the airport was finalized [6].

The Elbphilharmonie, a concert hall in Hanover, built from 2007 to 2016, is another well-known construction project due to its increased costs. At the beginning of planning, the opening was envisaged in 2010, whereby the total costs were estimated at around 186 million euros (EUR) [7]. Due to a multitude of poorly coordinated contracts and interfaces between executing companies, the project duration was delayed by about 6 years and led to construction costs of 789 million euros (EUR) [8].

The project Stuttgart 21, entailing the rearrangement of a railway junction in the city of Stuttgart, represents another example of the effects of inadequate planning and coordination. The scheduled project completion, initially planned for the end of 2019 to thebeginning of 2020, has been postponed several times. Meanwhile, the opening of Stuttgart's main station is scheduled for December 2025, and other parts of the project are scheduled for even later [9]. The costs of the project increased from 2.6 billion euros (EUR) to 9.15 billion euros (EUR) [7]. The extent of the variations in costs of the described projects can be illustrated in Figure 1.

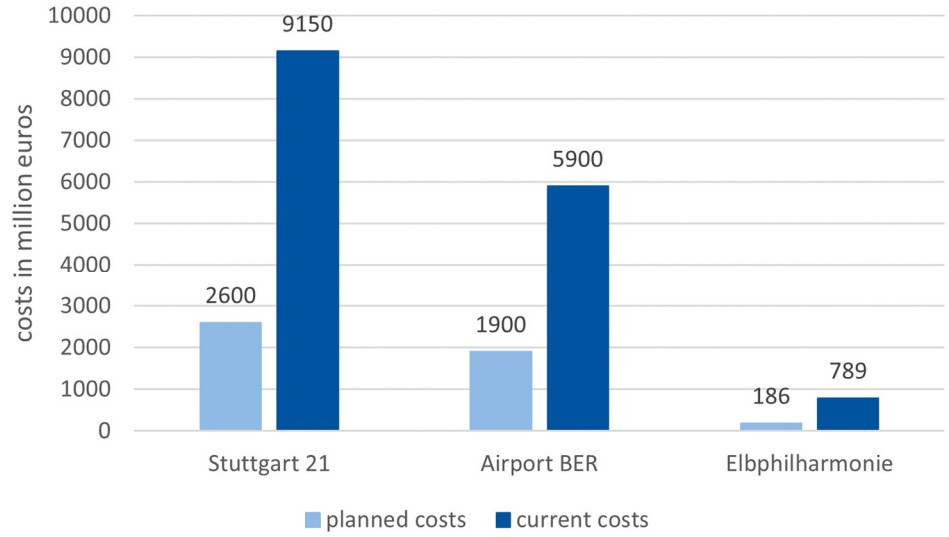

**Figure 1.** Planned and current costs of large-scale projects in Germany in 2017 (own illustration).

In Germany, the analyzed projects are classified in a special category for complex buildings, so-called "Sonderbauten", due to their size and type of use. Compared to residential and administrative buildings, projects in this category must fulfill particularly high requirements, e.g., in the area of fire protection. With increasing complexity of the regulatory and constructional requirements of a project, planning becomes more extensive with an increased risk of potential errors. As part of the critical infrastructure, hospitals in Germany also fall into the category of so-called "Sonderbauten" due to their special structural and regulatory requirements that need to be considered during planning and in further life cycle phases [10]. In addition, there is a multitude of use-specific requirements and stakeholders, such as doctors and the extra trade of medical technology.

The specific challenges in hospital construction can, for example, be presented by the construction project Centre for Operative Medicine 2 (ZOM II) in Düsseldorf. The project shows that hospital projects are prone to planning errors and associated execution deficiencies, mainly due to their high complexity. The ZOM II of the University Hospital Düsseldorf was mainly designed for surgically oriented hospitals. Five different clinics are integrated in ZOM II, including a neurosurgical clinic and an orthopaedic clinic. The clinics have a joint wing for surgeries with ten operating rooms. Furthermore, a central emergency department, a helicopter landing pad on the roof, as well as an intensive and intermediate care ward, are available [11]. For those patients who do not require intensive care, there are 288 beds in modern two-bed rooms [12]. Due to the organisation of these different subject-specific areas, the hospital requires distinct supply and logistics planning for the ongoing operational phase. The building of ZOM II was opened in 2014, with a delay of four years, and the additional costs amounted to about 180 million euros (EUR) [13]. University Hospital Düsseldorf has suffered a great loss due to the delayed commissioning. For example, equipment that had already been purchased could not be used and was already outdated at the time of commissioning. Additional burdens resulted from hiring new staff, which could not yet be employed due to the above-mentioned delays, but they, nevertheless, claimed their wages. The court proceedings, which have involved damages of 63 million euros (EUR), have not been concluded yet [14].

Effects of delays are particularly serious when construction work is carried out on existing buildings during the ongoing operational phase in hospitals. Construction delays can cause immense effects on the ongoing hospital operation and should be prevented to ensure the wellbeing and appropriate care of patients. In this context, the quality of the construction work should not be neglected under any circumstances, as otherwise harmful effects on the patient's health may occur (e.g., impure hospital hygiene). Furthermore, redundant protection of the functionality of vital equipment and devices in hospitals is necessary at all times to maintain supply in the event of technical malfunctions or emergencies.

It should be kept in mind that the medical field is in a constant process of change, which is reflected in even shorter innovation cycles in medical technology. This is accompanied by an increase in the necessary investment costs for the construction and operation of hospitals to be able to guarantee high-quality health care. The constantly changing medical, political, and constructional requirements often lead to hasty decisions. Therefore, hospital operators are unable to meet the demand with sustainable safeguarding. However, every renovation measure in a hospital leads to a strong restriction of the ongoing operation. The aim is to create efficient, sustainable, and goal-oriented planning by involving all necessary stakeholders at an early stage of planning to keep the costs, the effort, and the high complexity in hospital construction manageable. Thus, all relevant requirements should be considered from the very beginning, and interfaces can be defined in advance [15] (p. 3).

Renovation measures, extensions, or refurbishments can usually take place simultaneously with the ongoing operation of a hospital. By taking certain protective measures, hospitals can maintain their day-to-day operations. Therefore, the inconvenience caused

by the construction work, such as noise and soiling, must be curbed and adapted to the operational conditions [16].

### 1.2. Purpose

Due to these comprehensive effects, not only in terms of construction, but also in terms of social relevance, there is a huge need for action in the handling of project management in construction, according to Viering et al. in 2007 [17]. In complex buildings, the planning and definition of interfaces in the construction process play a vital role. Interfaces are points of contact between different matters or objects and are also defined in conventional linguistic usage as a connection or transition point between two processes or, in more general terms, between two areas. In construction practice, this consideration cannot be limited to two units, but it must be extended to a chain of units, which occur, among other things, in the form of tasks, activities, and responsibilities of the stakeholders.

Wukonig described, in 2011, how interfaces in construction projects arise not only between the stakeholders, but generally between functionally separated areas and tasks of a project [18] (p. 6f.). The points of contact arise in the subtasks of the different design work, the execution of different trades, as well as monitoring and control tasks. In many cases, this is due to the contractually demarcated areas of liability and responsibility. For coordinating interfaces, a proper and time-critical communication between all stakeholders is indispensable, starting with the definition of uniform terms. Otherwise, technical terms or concepts can be interpreted differently and cause confusion and misunderstandings. The aim of communication is primarily the exchange in information, as well as the identification of information gaps, to link the created subtasks with each other. In order to avoid diverging interpretations, there is a need to control the flow of information in a construction project so that information gaps are kept as narrow as possible. Control is achieved by defining the availability of data for stakeholders to avoid contradictory information and to ensure that the flow of information does not become excessive, as well as that the actual activities are not neglected. Further control elements are available in the form of checklists, tables, or as plans with standardised presentation rules [18] (p. 78f.).

In 2015, Lin developed the ConBIM-Interface Management system to enhance interface information sharing and tracking efficiency. It uses three-dimensional interface maps and the BIM approach to track and manage interfaces in a graphic form [19]. In 2018, Alaloul et al. presented potential benefits, as well as challenges and strategies, for the introduction of smart technologies in the construction process [20]. Whereas, in 2022, Safikhani et al.'s paper highlighted the concrete benefits to the construction process of using VR technology in combination with the digital method Building Information Modeling (BIM) [21]. In 2022, Sun and Liu proposed a novel hybrid model of Digital Twin-Building Information Modeling. The model helps in assisting the dispatch systems in the construction projects to a greater extent than when compared to the implementation of individual technologies [22]. In an article from 2022, Li et al. proposed the novel idea of Hospital Information Modeling, an expansion of BIM for hospital settings [23]. Sepasgozar et al. presented a roadmap for developing and implementing disruptive technologies for the construction industry, in 2023 [24]. Various digital technologies, such as BIM or augmented reality (AR), are presented to optimize processes.

In this article, the process of construction measures is itemised regarding their problems concerning the reduction of the effects of planning and construction faults in the future. These difficulties are identified and analyzed using the example of hospital construction in Germany, as there is a particularly large number of interfaces to be defined due to the special characteristics of the use and requirements arising from critical infrastructure. Based on this, solution approaches are developed to optimize these interfaces and to reduce mistakes in the planning phase. In developing the solutions, digital methods are used to simplify the flow of information and communication and, thus, to intervene at the source of the interface problems. The solution approaches are transferable to other complex buildings with infrastructural, as well as social, significance.

## 2. Methodology

The literature research was carried out to obtain information about the status quo regarding interfaces in construction process and possibilities for optimization using digital methods for describing the research gap and the purpose of this article. The issue was dealt with using the example of hospital construction, since, due to the complexity caused by a higher number of trades, stakeholders require more interfaces than in other types of building. Since this work was developed as part of the research project KlinikBIM, whereby a guideline for the implementation of BIM in hospital construction in Germany is being created, the requirements for hospital construction in Germany were taken into account. In order to obtain background information, including the legal framework, as well as the basic requirements of construction measures in hospitals, a review of the literature was carried out.

Based on these findings, an analysis of interfaces in hospital construction processes was conducted through ten guided interviews with experts on the topic, who are from Germany. They are also part of the KlinikBIM project consortium. The interviewed experts are experienced in the field of hospital construction and belong to different subject-specific areas, such as structural design, architecture, or the construction and operations department of the operator. Thereby, the intention was to identify interfaces in hospital construction, as well as to obtain an opinion on digitization topics and their potential for optimization. The interviews were conducted by video conference with the use of a guideline. Although the questions were specified, the interviewees were still able to answer freely and give examples. The interviewer was also given the opportunity to ask questions in this regard. Mayring's qualitative content analysis was chosen for the evaluation of the guided interviews, which aim to filter out a certain structure from the material to apply it in the form of a category system.

The guided expert interviews and qualitative content evaluation offer the advantage of generating new insights while proceeding in a highly rule-governed and schematic manner. Furthermore, the described methodology enables the establishment new theoretical considerations, based on interview transcripts. Because of the underlying transparency of qualitative content analysis and its applicability to a wide variety of content, it represents an adequate tool for gaining scientific knowledge in the present case.

The evaluation method allows us to assign and to analyze the different interfaces to superordinate problems, such as errors in the planning or in the construction phase. Based on the results from the literature research and the interview evaluation, solution approaches for optimizing the interfaces could be developed. In addition, strategic recommendations for action are given.

## 3. Legal Background

In this research report, the legal requirements of the health care system and building law are exemplified by the German regulatory framework. Due to the federal structure of Germany, the building regulations differ in the various German states. Nevertheless, the deviations and the influence on the objective of the present research can be assessed as minor.

In Europe, the primary approach is the sovereign regulation of buildings by the state. There is a superordinate European framework of building regulations, such as EU Directive 305/2011, which regulates the marketing of construction products in the European Economic Area and defines the basic requirements for buildings as a European standard. Due to the principle of subsidiarity prevailing in the EU, the implementation of these requirements is the responsibility of the EU member states themselves. On this basis and the European interest in the exchange in goods and products, most European countries have building regulations and standards that may differ in content, but they serve to avert hazards and to ensure a functioning coexistence [25] (pp. 11 and 16).

In this article, Germany/Lower Saxony's ordinances and laws were used as an example, as this work is connected with the research project KlinikBIM, and the experts of the consortium were from Germany, predominantly from the federal state of Lower Saxony.

Besides new constructions, measures to secure and further develop existing buildings, such as reconstruction and renovation measures, were also considered. This leads to a large number of laws and regulations, which partly need to be taken into account in hospital construction and partly can be regarded as recommendations. However, since there is no clear nationwide model ordinance on the requirements for hospitals, some federal states, such as Baden-Württemberg or Brandenburg, have issued recommendations and ordinances. On the contrary, Lower Saxony follows the model hospital ordinance or works on the basis of the recommendations and ordinances of other federal states because the model hospital ordinance only serves as a guide and is thus formulated with restraint [10].

In addition to the building regulations, there is a particular fee schedule in Germany, which represents the chronological sequence of a planning process by means of service phases [26] (p. 96). In addition, the scope of services for project management is described by the German Committee of the Associations and Chambers of Engineers and Architects for the Fee Schedule (AHO) in the AHO publication series in booklet no. 9 [27]. The descriptions of the AHO and the fee schedule are not statutory requirements, but merely a standard established in the market. Furthermore, there are numerous standards that are relevant to hospital construction, for example, DIN 1946-4 for ventilation and air-conditioning systems in the healthcare sector [28], DIN 6812 for medical X-ray systems [29], or DIN EN ISO 7396-1 concerning piping systems for medical compressed gases and vacuums [30].

## 4. Findings from Interviews to Identify the Key Interfaces in Hospital Construction

To analyze the existing interface problems in hospital construction, the expert interviews were considered and evaluated separately. Based on the resulting findings, the interface problems can be narrowed down and analyzed. For all interface issues, the later collisions are identified, and their impact is greater. Consequently, the following evaluation is used to develop solution-based approaches that can be used to facilitate construction measures in the future.

### 4.1. Identification of Important Stakeholders in Hospital Construction Measures

According to the interviewed experts, the hospital is considered the building owner. Often, the hospital has its own construction department, which is responsible for planning and construction measures in the building. It represents the interface between the hospital executive board, the supervisory board, the board of directors, the physicians, and the nursing staff. Furthermore, the interests of the patients and their visitors must be taken into account. Regarding the building law, building supervision authority is the highest priority. Furthermore, municipalities, as well as representatives of public interests, participate in target planning at the state level, and they support the administrative level. They ensure that the interests of the municipality are respected, as well as that development plans are created and adapted, if necessary. Furthermore, other authorities involved, such as the nature conservation authority or the trade control, belong to the administrative level as representatives of public interests. Another representative of public interests is the fire brigade, which must be involved in fire protection measures.

The realization of the building project begins with the commissioning of the planning services. Especially, for new hospital buildings, general planners are commonly commissioned to simplify communication. However, working with architects and various specialist planners is also feasible. Due to the complexity of the planning, a consultant advises the building owner. The large number of specialized planners led to the commissioning of the executing companies, which is usually carried out as an individual contract. To assist the owner, a project manager takes over the coordination of the different stakeholders. Legal consulting can be considered for general advice or for decisive questions. The most

important stakeholders in hospital construction, according to the interviewed experts, are outlined in Figure 2.

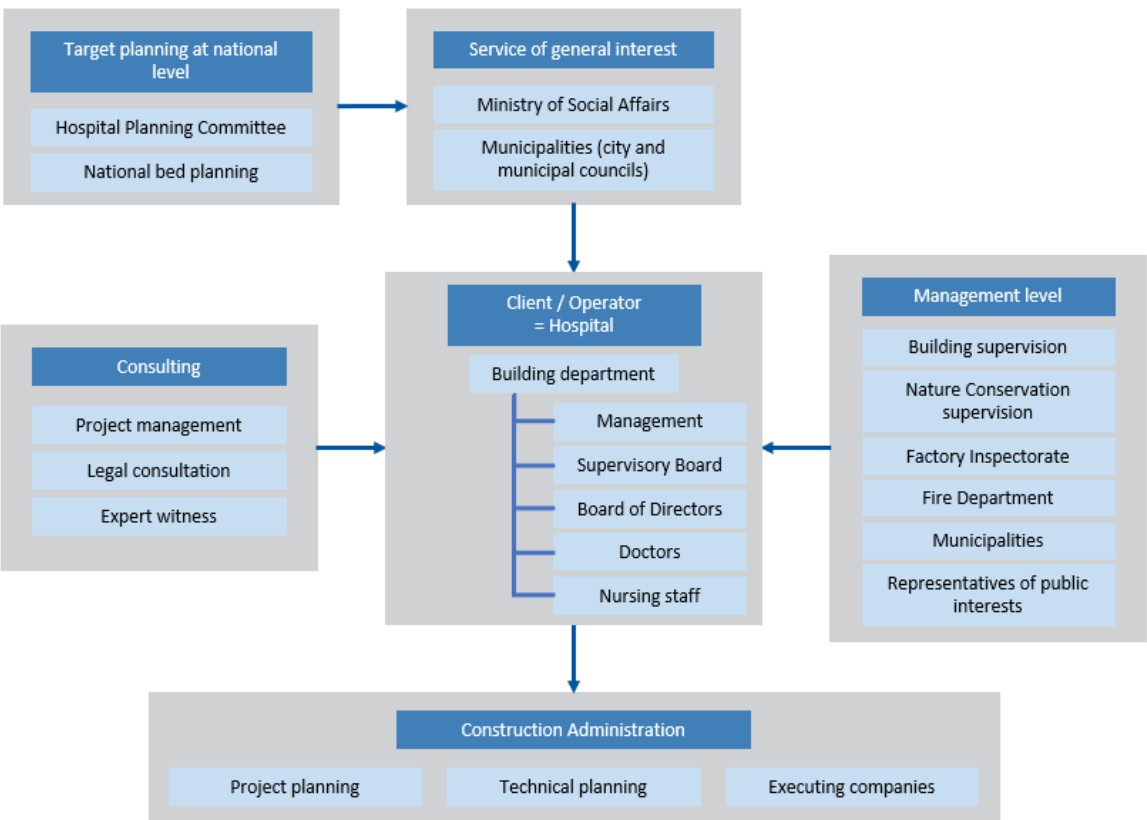

**Figure 2.** Stakeholders in clinic constructions (own illustration).

### 4.2. Main Building Trades and Interfaces in Hospital Construction

This section focuses on the various building trades and disciplines that need to be taken into account in hospital construction to avoid collisions and interface issues, according to the experts. Besides the typical collisions of the building trades of heating, ventilation and air conditioning (HVAC), as well as sanitary services and electricity, the experts have reported on many other interfaces in hospital buildings, which occur in view of high safety requirements of fire protection and hospital hygiene. Furthermore, medical and laboratory technology represent another challenge in terms of early and precise planning. In this context, it is important to ensure that reserves are planned so that dimensions and ports suffice in the event of change or replacement in equipment. Electrical engineering is one of the largest building trades involved, which requires precise planning of cable routing in coordination with the technical building equipment. A list of the most important disciplines is shown in Figure 3.

Based on the evaluation of the interviews, the main interfaces of measures in hospital construction are summarised in Figure 4, as follows.

### 4.3. Impact of Interfaces Using the Example of Doors

According to the experts, decisive components with high coordination efforts are the doors located in the corridors. Due to the requirements of the necessary functionality in running a hospital, these are an essential element and example of the undisturbed operational flow. Doors have high demands on fire protection and access restrictions and need to be planned and executed by different trades. For electrically operated doors, this includes the door itself, the supply line, as well as the internal wiring for controlling the

door, provided by electrical engineering. An exemplary structure of an automated door system with access restriction is shown in Figure 5.

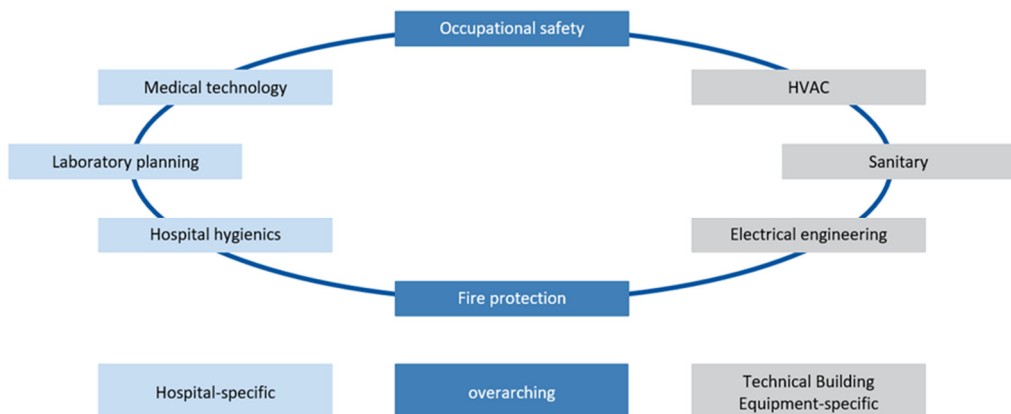

**Figure 3.** Integral technical planning in hospital construction (own illustration).

| | Heating, ventilation, air conditioning | Sanitary | Electrical engineering | Medical technology | Fire protection | Architecture | Doors |
|---|---|---|---|---|---|---|---|
| Heating, ventilation, air conditioning | | X | X | X | X | X | |
| Sanitary | X | | X | X | X | X | |
| Electrical engineering | X | X | | X | X | X | x |
| Medical technology | X | X | X | | X | X | |
| Fire protection | X | X | X | X | | X | x |
| Architecture | X | X | X | X | X | | x |
| Doors | | | x | | x | x | |

**Figure 4.** Evaluation of the main interfaces and collisions, which are marked with an "×" (own illustration).

Due to the large number of planners and trades involved, this is where most of the interfaces arise, and particular attention is needed during the planning phase. According to the experts, in conventional planning, some services, such as the supply line of the doors, are forgotten to be tendered, which is only noticed during execution. Consequently, a supplement is issued, which considerably increases the additional costs due to the large number of doors in a hospital.

Another difficulty is that doors are usually components with high fire protection relevance. Doors along escape routes must be able to be opened manually in the event of

fire, but they also protect against the spread of smoke in the building. Moreover, automatic opening is often required for hospital operations because patient transport in beds is part of everyday operations, especially in the corridors. Furthermore, it is necessary to consider functional areas, which may only be entered by personnel using access authorization. In this area, doors must be installed in a manner that allows them to be opened in the direction of the escape route at any time, even without the electrical presentation of authorization. In the event of fire, the doors must be able to be opened from both sides without the electrical presentation of authorization. For this reason, the wiring and the direction of installation must be considered during installation. Card readers for checking access authorization need to be installed in the immediate vicinity on the correct side and connected to the door.

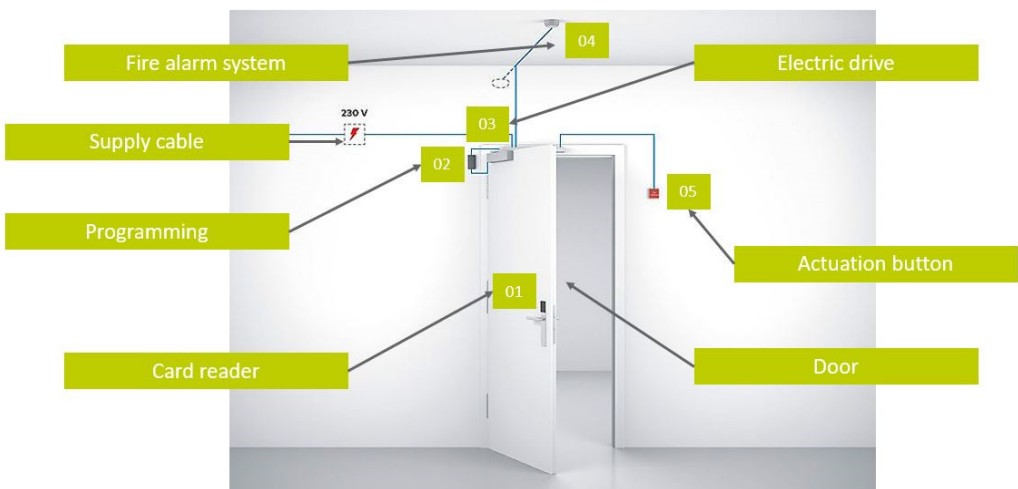

**Figure 5.** Structure of an automated door system with access restriction. (Adapted with permission from [31]. 2020, Dormakaba Group.)

## 5. Solution Approaches

### 5.1. Application of Digital Methods

The implementation of digitalization in the projects of the interviewed experts varies. On the basis of the expert interviews conducted, assuming that interfaces can be reduced by means of digital planning, it was analyzed whether the options listed below can lead to a reduction in interface problems.

#### 5.1.1. Impact by Application of BIM

The BIM method is considered a pioneer of digital planning. In terms of hospital construction and the many different and complex interfaces, BIM is a suitable tool for reducing interface problems. If applied consistently, a BIM model contains all project information and keeps the data centrally in a browser- or cloud-based project space, a so-called Common Data Environment (CDE). The information attached to the individual building components simplifies the calculation of quantities and costs, e.g., by determining the number of necessary fire compartments directly from the model. This is a significant advantage, especially in hospital construction, due to the complex requirements for ventilation technology. It also simplifies the exchange in data between the stakeholders and ensures that everyone is on the same planning level to minimize the number of design errors due to a lack of communication. As a result, problematic interfaces and collisions can be identified and rectified in the early stages of planning. In a further part of this article, it is described how earlier handling of planning errors has a lower impact on costs than if it would be remedied at a later stage of planning (Figure 6).

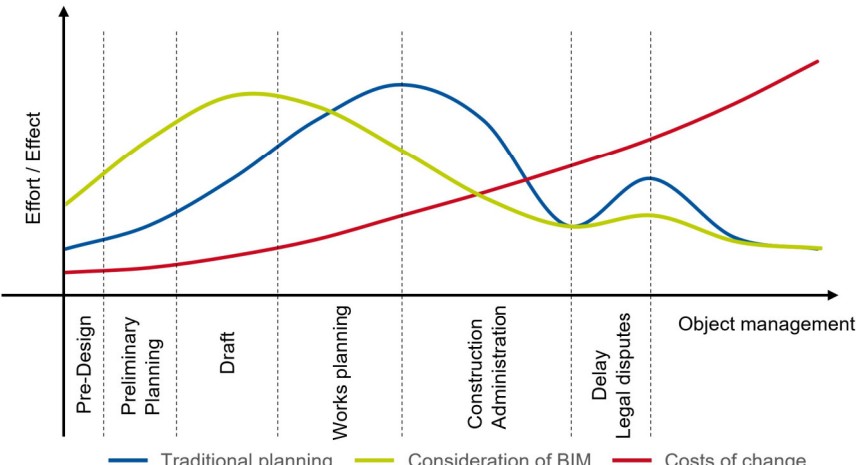

**Figure 6.** Effort of traditional planning and planning with BIM, according to the MacLeamy curve (own illustration based on [32] (p. 32)).

BIM also offers enormous advantages for facility management. With a digital twin, an economical and effective operation in a hospital can be facilitated. The main advantage of a digital twin can be seen in the aspect of a central data source to which everyone involved has access. With the right management, all necessary data are always available and up-to-date, so that the necessary information is available at all times in the event of repairs, maintenance, and servicing measures. However, continuous adaptation and updating of the model is necessary, already, in the operational phase. Moreover, a BIM model can only be used successfully under the condition of fixed collision rules and project-related definitions in close consultation with all stakeholders. This is performed through the elaboration of the so-called Employer's Information Requirements and the BIM Execution Planning, in which the exact procedure for using BIM in the project is described. However, these procedures are out of the scope of the present paper.

The implementation of the BIM method in Germany entails several obstacles, as revealed in the interviews. One significant problem is the lack of specifications that define the remuneration and costs of BIM services. It should also be considered that procurement costs are initially incurred for IT, as well as for the qualification of staff to use BIM. Accordingly, some hospital operators have been rather critical about the implementation of BIM so far. Furthermore, it was criticized that the actors involved in the planning process assume a higher than necessary planning effort. In contrast to the conventional method, there is no increased total planning effort with BIM, but efforts are shifted to earlier sub-phases before execution (see Figure 6). By shifting the effort into earlier project phases, the costs of unforeseen planning changes will be kept lower compared to later changes because the gradient of the cost of changes increases during the progress of the project.

Despite the positive influence of BIM on the coordination of construction projects, disruptions, such as delays due to delivery bottlenecks or unforeseen events, are still possible. Furthermore, the use of BIM creates new digital interfaces, which must also be coordinated in the already mentioned Employer's Information Requirements and the BIM Execution Planning. In the event of occurring planning errors, the stakeholders often refuse responsibility, which ultimately leads to legal disputes. These disputes can only be reduced by clarifying responsibilities at an early stage, e.g., by applying the BIM methodology.

### 5.1.2. BIM Use Cases

In the further course of this chapter, further digital use cases are described in order to reduce interface errors, whereby BIM constitutes a prerequisite for application. By means of checking design variants, different technical concepts are examined at the beginning of the planning regarding the investment costs. Thereby, functional influences are focused and

constitute the basis of the examination for selection of the most economical variant. This is carried out in the early planning stage to determine the demand and is expected to result in an increased degree of cost and schedule certainty, as well as faster commissioning. In addition, it is possible to efficiently design and optimize the routes between the functional areas using a variant check. Operational concepts of, for instance, fire protection officers or hospital hygienists, are to be examined individually by the various professional actors regarding their feasibility and economic efficiency. Furthermore, the functional plans and designs need to be checked for compliance with current regulations in order to identify collisions or planning errors before the start of construction works.

So far, according to the interviewees, a major part of the variant checks is still carried out based on two-dimensional plans. Due to the high number of plans and concepts, as well as the complex planning requirements, it can be assumed that errors may be overlooked, despite a variant check, leading to collisions during execution. A possibility for improvement is the application of the BIM methodology, which ensures that there is only one model with all designs and information to check.

Another possibility to detect planning errors is to carry out simulations during the planning phase. The most common type of simulation involves energy considerations, e.g., to specify heating and cooling cycles. Simulations can also improve the efficiency of the planning of ventilation systems. By visualizing the architectural geometries, pipe routing can be planned and checked for economic efficiency. Another advantage of testing different geometries stands out in fire protection planning. In this way, smoke development in the event of a fire can be simulated and contained with suitable measures. If a preliminary check takes place by means of simulations, it is possible to simulate routes of insertion for medical equipment, as well as of technical systems to eliminate possible complications during planning and to adjust the dimensions. In this context, the simulative preliminary test of the so-called "active principle test", which is relevant in Germany under building regulations, is referred to. It is used to test the interaction of different technical systems in the event of a hazard.

However, up until now, simulations are often carried out with conventional software due to a lack of a BIM implementation strategy. By using the BIM methodology, there are further possibilities, in addition to the advantages mentioned above, to improve planning and construction site work processes and to represent them in an enhanced degree of detail. Virtual reality (VR) can be used to create computer-generated, interactive environments that can be displayed in real time with terminal devices, such as VR glasses, tablets, or smartphones at no great financial expense. The combination of an existing three-dimensional model with VR enables variant considerations, preliminary tests, and collisions to be displayed and checked more clearly. Special attention is paid to the flow of logistical processes. In this way, fire protection concepts, escape routes, traffic routes, or even logistical effects of construction measures can be visualized and adapted in advance.

Starting with fire protection concepts and the planning of escape routes, simulation, in combination with VR, can be useful. In the past, fire protection measures were only tested in their technical function in order to be able to exclude functional errors in the equipment. By visual inspection of the building, escape routes can be viewed in their entirety and checked regarding their stability in the event of a fire. The use of VR also provides a visualization of the development of fire and smoke, which makes necessary adjustments to the fire protection systems visible.

In addition, the traffic and transport routes must fulfill a multitude of constructional requirements. Due to the high number of these routes in a hospital, a manual check is unfeasible. Using a virtual passageway, routes, door widths, room heights, area requirements, lighting, and markings can be checked. Especially, door and corridor widths must be checked with regard to bed transport or accessibility. The introduction of new medical equipment can also be simulated at an early stage using VR, so that problems with insufficient door widths can be rectified before delivery. VR can be also used to visualize

the planning for laypersons so that hospital specialists can check whether all necessary connections and equipment have been taken into account.

Construction measures during ongoing hospital operations always have an impact on the supply and functioning of a hospital. To keep these effects to a minimum, construction measures are demarcated with dust protection walls. By using virtual simulations, the effects can be determined in advance so that measures can be taken to protect patients in advance. In summary, the use of VR aims to optimize processes and to eliminate planning errors and collisions, with minimal effort at the beginning of planning, to avoid delays, to reduce additional costs, and to enforce efficient and comprehensible workflows.

While VR completely replaces reality with virtual sensory impressions, AR aims at a computer-supported extension of reality perception, where the view of the real environment is supplemented with computer-generated superimpositions. By overlaying virtual and real points, visual information, the location of components, and the positioning of reference points on the building, construction experts and laypersons can use visualization to initiate potential changes. The application of AR is not specific to a project phase and can be implemented throughout the entire building life cycle. During the planning phase, the use of AR allows representatives of different trades to view the BIM model from different perspectives. Models can be analyzed more efficiently, and potential problems can be identified by showing and hiding single units. Furthermore, the superimposition of design models, such as a technical building equipment model, with reality, enables early detection of conflict points, which can be rectified before construction work begins.

It is further possible to check and record the current status of construction by overlaying planning models with scan models of the building object. In addition, changes and construction deviations can be checked and imported back into the model to obtain an as-built model at the end of execution. In addition, the use of AR offers the option of projecting plans onto building components, e.g., in order to be able to show pipe to avoid damage during construction. Additionally, it may prove to be useful to prevent execution errors due to the attachment of components or the preparation of breakthroughs with the help of the exact visual specification.

In addition to the advantages of AR in the planning and execution phase, the possibility of virtual representation of the as-built model serves to simplify maintenance, repairs, and servicing measures in the operational phase. By feeding back the manufacturer information of maintenance-relevant components into the BIM model during the construction phase, an automatic maintenance report of the objects can be created, which reduces the workload of facility management. Maintenance technicians can then be sent to the corresponding object and quickly identify it using indoor navigation via AR glasses. This creates huge potential for reducing maintenance working hours by linking virtual step-by-step instructions to the maintenance-relevant components. In the future, there is the possibility of being able to practice standardized work steps by combining them with artificial intelligence and, thus, reducing costs with the help of standardized and routinized processes.

The digital applications mentioned above serve the communication of the project participants across the entire life cycle. Using the virtual models, location-independent meetings with a view of the construction project can be realized. Thereby, cross-trade problems can be discussed rapidly without tedious coordination effort. In addition, the use of visual forms of representation is recommended regarding visits to authorities, as well as citizens' initiatives. By overlaying the construction status with relevant information from the BIM model, use cases can be viewed together on the construction site or in the office. Particularly, with regard to the public purchasers in hospital construction, as well as the many complex requirements and conditions, the possibility of an uncomplicated joint consideration of critical points is enforced. Furthermore, due to the many necessary agreements with the various parties involved and users from other disciplines, a virtual view of the planning represents a great benefit in order to view the interactions of emerging projects in their environment and to contribute to the design.

In summary, with digital methods, such as BIM, many interface problems, such as geometric collisions, can be minimized. However, interdisciplinary tasks for the execution of construction are, at this point, difficult to coordinate.

*5.2. Development of a Task and Trade Control Matrix*

Due to the high level of complexity and the large number of stakeholders, but also for user-specific coordination adapted to the building, there is a need for technical and contextual coordination. Based on the findings from the expert interviews, a trade control matrix is considered a suitable tool, especially for hospital construction. Within the framework of the present study, a first draft of such a matrix was developed, which is illustrated compactly with reference to relevant information.

Up until now, control matrices have only been used for safety-related functions and their functional correlation, e.g., as a basis for the active principle test of technical equipment. The intention is to assist specialist planners, building owners, and operators, as well as executing companies, in keeping an overview and being able to present hazard scenarios. At the beginning of the planning process, a risk analysis is performed to determine possible and probable hazards in terms of their use and environment. The risk assessment must be continuously reviewed and updated throughout the entire life cycle of a building.

For the purpose of this paper, the principle of safety control matrix is transferred to the entire construction project and its stakeholders for an optimization of planning and execution. Thereby, the control matrix should be applied and updated by all stakeholders in each project phase. The associated advantages are demonstrated through clear definition of the interfaces and the corresponding responsibilities, which facilitate coordination and reduce errors in planning and execution. The precise demarcation of interfaces clarifies who is responsible for and involved in which part of the construction measures from the beginning. Nevertheless, should errors occur, those responsible can be clearly identified on the basis of the task control matrix. Another advantage of using a task and trade control matrix becomes apparent in construction supervision. Site managers and project controllers obtain an improved overview, with all the necessary information compactly summarized and organized. Furthermore, executing companies obtain more insight into their areas of responsibility and can schedule their work more easily.

A task and trade control matrix was created with specification of the building trades involved in hospital construction (compare Figure 7). In this context, it should be noted that only the main building trades are listed in this figure for simplification. As participants vary in every project, they need to be adapted specifically. In addition, heating, ventilation, and air conditioning are grouped together for a simplified illustration; likewise, the executing trades in building construction have been kept general and are to be adapted to the corresponding building component.

By specifying the trades involved and naming the subcontractors and their contact persons, an overview of the distribution of tasks and responsibilities can be created. Specifying the location of the components helps to further enable a direct assignment of the corresponding component. In this way, the division of tasks for a construction service can be applied to a large number of identical components. The matrix merely needs to be multiplied and related to its respective component. Subordinately, the construction work can be divided into its sub-services, so that an exemplary checklist for an automated door system with access restriction is created based on the fundamental requirements elaborated through the literature research and the expert interviews (see Figure 7).

**Construction:** Automated door system with access restriction

| Location component | Building section A | | | | | | ... |
|---|---|---|---|---|---|---|---|
| | 1. Floor Corridor-1 | 2. Floor Corridor-1 | 3. Floor Corridor-1 | 1. Floor Corridor-2 | 2. Floor Corridor-2 | 3. Floor Corridor-2 | ... |

X = Responsibility
I = Indication
C = Coordination
S = Support

| Task | Comment | Client / User | Technical Building Equipment - HVAC | Technical Building Equipment - Sanitary | Technical Building Equipment - Electricity | Building construction | Fire protection | Medical technology | Hospital hyginics | Occupational safety | Information (references, standards, plans, documents, etc.) |
|---|---|---|---|---|---|---|---|---|---|---|---|
| **Planning** | | | | | | | | | | | |
| Selection of systems | | X | | | | | | | | | |
| **Supply cable** | | | | | | | | | | | |
| Electric drive | | | | | X | | | | | | |
| Actuation button | | | | | X | | | | | | |
| Card reader | | | | | X | | | | | | |
| Fire alarm system | Cable 230V | | | | X | | C | | | | |
| **Delivery and installation** | | | | | | | | | | | |
| Door | Opening in direction of escape | | | | | X | | | | | |
| Electronic cylinder | | | | | X | | | | | | |
| Actuation button | | | | | X | | | | | | |
| Card reader on the wall | | | | | | | | | | | |
| Fire alarm system | | | | | X | | C | | | | |
| **Wiring** | | | | | | | | | | | |
| Electric drive | | | | | X | | | | | | |
| Actuation button | | | | | X | | | | | | |
| Card reader | | | | | X | | | | | | |
| Fire alarm system | | | | | X | | C | | | | |
| **Programming** | | | | | | | | | | | |
| Closing forces and speed | | | | | X | | | | | I | |
| Initial coding closing element | | | | | X | | | | | | |
| Access restriction | | I | | | X | | | | | | |
| Fire alarm system | Automatic unlocking by alarm or power failure | | | | X | | C | | | | |
| **Commissioning** | | | | | | | | | | | |
| Commissioning door | | | | | X | | | | | | |
| Signage | Note access restriction | | | | | | | | | X | |

**Figure 7.** Exemplary task and trade control matrix using the example of an automated door system with access restriction (own illustration).

Ultimately, the approach of a task and trade control matrix pursues the intention of developing a standardized processes for construction services and creating generally applicable interface catalogues that can be used for future construction projects. Using this task and trade control matrix, interfaces can be delineated at an early stage, thus reducing possible planning errors. By linking it to a construction schedule, supplements, disputes, and the resulting extensions to construction times to complete construction measures are reduced. In addition, it is possible to link the component-specific task and trade control matrix directly to the respective objects in the BIM model in order to visualize the state of execution. By programming the matrix of a component, it can be linked to the same component of the object and catalogued in the BIM model and identified by its location. By selecting the location of the respective component in the matrix, the correct object is highlighted in the model. The query about the status of the construction is then made based on the matrix and can be displayed in the BIM model. Thereby, a laborious search for the appropriate component in the model can be avoided, and the status of the construction work can be queried individually for each component.

Lastly, the virtual representation in the BIM model makes subtasks clearer. If this option is further combined with the possibility of VR / AR, the components can be viewed directly on the construction site or in the office. Thereby, responsibilities can be better represented and more easily documented in meetings with the stakeholders. Additionally, transparency is created regarding the construction work and its subtasks, by which the stakeholders have both a written description of the work and a visual image from every perspective. As a result, ambiguities can be minimized, and execution errors can be avoided.

## 6. Conclusions

The planning and execution of hospital buildings is highly complex due to the numerous requirements described. Using BIM in connection with a CDE, communication can be improved and become more transparent, which is indispensable for complex construction projects. More efficient results can be achieved through a regulated flow of information regarding the timeline and content. As a prerequisite for the application of BIM, all stakeholders need to possess the necessary software programmes and have the appropriate software knowledge and basic knowledge of the BIM application. Furthermore, the project participants should use an exchangeable format, such as Industry Foundation Classes (IFC), to fully exploit the potential for communication. Accordingly, the openBIM approach is explicitly recommended for handling.

Furthermore, the use of BIM as the basis for processing of integral building planning enables recognition and elimination of interfaces and collisions at an early stage. Due to model-based data, BIM also provides the opportunity to carry out simulations, such as the spread of fire and smoke or the evacuation of a building, to perform necessary changes early on. After all, the later interface problems are identified, the greater the impact, since the cost of change increases exponentially over the course of the project. Planning errors can be identified through automated model-based checks with BIM. However, for the identification of interdisciplinary interface errors in construction, the use of a task and trade control matrix is recommended, in which the interfaces and responsibilities are demarcated, leading to a decrease in the potential for interface errors. The intention is to link BIM models to the matrices, so that they can be assigned to the respective objects in the digital model. Although the solutions presented in this report were created based on hospital construction, they can be transferred to any other building type. Especially complex projects can benefit from this approach, as the described challenges and risks are particularly important to clearly define the interfaces. Nevertheless, an adaptation of the tasks and trades in the matrix is necessary for different building types and for each project. Furthermore, it is important to mention that the developed solution approach in this present study has not been piloted. In future, the application of the task and trade control matrix in connection

with BIM needs to be examined in a pilot project, and the results need to be reflected and, if necessary, adapted. The benefits and effort are to be analyzed.

Notably, the German regulations are not yet adapted to the application of the BIM methodology. Since the introduction of the BIM method cannot be simplified by adapting a single set of rules, a considerable amount of adaptation of the legal framework in the construction industry is required. Besides the fee schedule, public procurement law, and contract design, the question of liability needs to be clarified. Especially, in hospital construction, many legal regulations must be taken into account, which are interrelated. An additional complicating factor in Germany is that many requirements must be observed and laid down in different laws and regulations, and, thus, they differ partly in the various regions of Germany. Therefore, based on the results of this work, the need for a specialized digital directive in hospital construction for national application is expressed. This should simplify the construction process regarding the merging of hospital construction guidelines and ancillary rights and present them more clearly.

To ensure the optimization of construction projects in hospitals by using BIM, strategic change management must be established in the construction departments of the hospitals in order to qualify the staff accordingly. To spread and promote the use of BIM in hospital construction and to maximally exploit the advantages of BIM, clear recommendations for the use of BIM should be formulated. Accordingly, an application-oriented guideline for the step-by-step implementation of BIM in hospital construction projects is required. The guideline should describe an adequate BIM strategy with potential use cases based on the complex requirements and framework conditions in healthcare construction. To successfully advance digitalization in the construction industry, it is vital to adapt and further develop various digital supplementary tools, e.g., for model-based simulation or the application of VR and AR. In regard to the entire life cycle of buildings, relevant data from the as-built model must be linked to the facility management software environment by digital tools. Considering that existing buildings are not designed with BIM and that there is no digital twin, difficulties emerge in the planning of reconstruction. Due to outdated or missing plans, construction measures are difficult to arrange without collisions and plan changes occurring during construction. In this respect, the elaborated solution-approaches are not always applicable to conversion measures. Further research is needed to determine the economic impact and worth of digitally scanning existing buildings for the operation phase. Thereby, it can be decided whether and in which cases digital methods are useful for renovations and extensions of existing buildings. A comparison of the economic aspects in relation to the effort involved can help to balance the disadvantages and benefits.

**Author Contributions:** Conceptualization, S.H., D.G. and S.K.; methodology, S.H., D.G. and S.K.; validation, S.H., D.G. and K.K.-A.; formal analysis, S.H., D.G. and S.K.; investigation, S.H. and S.K.; resources, D.G. and S.K.; data curation, S.H., D.G. and S.K.; writing—original draft preparation, S.H. and S.K.; writing—review and editing, S.H. and K.K.-A.; visualization, S.K.; supervision, K.K.-A.; project administration, S.H.; funding acquisition, K.K.-A. All authors have read and agreed to the published version of the manuscript.

**Funding:** This research paper was written as part of the project KlinikBIM. This project was funded by the German Federal Institute for Research on Building, Urban Affairs and Spatial Development on behalf of the German Federal Ministry of the Interior, for Building and Home Affairs with funds from the Zukunft Bau research promotion.

**Data Availability Statement:** The transcribed interviews are not publicly available due to privacy restrictions.

**Conflicts of Interest:** The authors declare no conflict of interest.

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
