# Peer review of "Optimizing Interfaces of Construction Processes by Digitalization Using the Example of Hospital Construction in Germany"

_buildings, doi:10.3390/buildings13061421_

Round 1

Reviewer 1 Report

Dear Authors,

The Manuscript ID buildings-2325519, generally presented good quality research material in highlighting, the hypothesis, and methodological statement. The selected problem of the topic “Optimizing Interfaces of Construction Processes by Digitalization” is interesting. The point of the view on to analysed of the problem and the author’s position is interesting. Several comments regarding the improvement of the manuscript are following:

The presented title of the manuscript presented an example of Hospital Construction, but in the main text presented example the hospital construction example is missing. A practical example of doors is presented (page 8, line 321-page 9, line 344). Regarding this, the title of the manuscript must be changed.

The literature analysis from the selected problem of the topic with a new literature source (from the 2022-2023 year) from the scientific journals can be more analysed. 

The presented methodology is not sufficient.

The framework with a digitalization process of the presented example can be presented in the manuscript.

In the manuscript review part, a higher number of the literature source from the journals can be presented. The presented review part of the manuscript and list of the presented literature reference can be improved.

A comparison of the  results can be presented.

The main text on page 15 lines 594-603 must be presented in the English language.

Reviewer

Author Response

Dear Reviewer,

thanks a lot for your review and comments to improve the article. In this cover letter, we list and explain the points of revision based on your comments. Furthermore all revisions to the manuscript are marked up can be tracked in the Word file.

  • The presented title of the manuscript presented an example of Hospital Construction, but in the main text presented example the hospital construction example is missing. A practical example of doors is presented (page 8, line 321-page 9, line 344). Regarding this, the title of the manuscript must be changed.

After consultation with the co-authors, we would prefer to leave the title as it is. Since hospital construction is very error-prone with regard to interfaces, due to the circumstances described, we think this example makes sense. The literature research already refers to hospital construction and lists the example from Düsseldorf. Since the interviews are also conducted with experts from the hospital sector and the task and control matrix is created using the example of hospital construction, we find the title appropriate and I hope we could convince you.

  • The literature analysis from the selected problem of the topic with a new literature source (from the 2022-2023 year) from the scientific journals can be more analysed.

We explicitly searched again for current literature on the topic, but could not find anything current. This was noted in the description of the research gap. Nevertheless, we have added two more studies, but they are not more recent than the ones mentioned so far.

  • The presented methodology is not sufficient.

Has been edited.

  • The framework with a digitalization process of the presented example can be presented in the manuscript.

The prerequisites for a digital work process are described in the conclusion. If desired, I can also add them already in the chapter of results.

  • In the manuscript review part, a higher number of the literature source from the journals can be presented. The presented review part of the manuscript and list of the presented literature reference can be improved.

Two more studies have been added.

  • A comparison of the results can be presented.

In the conclusion, we address the limitations of our study / results. For example, that the matrix, which was developed using the example of hospital construction, can also be applied to other types of buildings. However, we point out that in the future the solution approach must be piloted on a practical project, since this did not take place in the context of this study.

  • The main text on page 15 lines 594-603 must be presented in the English language.

Has been edited. In addition, the English has been optimized in some places.

Best regards,
Sabine Hartmann

Reviewer 2 Report

Dear author,

Thank you for your effort in producing this article.

I have reviewed your article and have some suggestions for improvement:

1. Please revise your affiliation because there is some missing information. For instance, postcode and country.

2. Please add to your abstract, mention the major findings of your paper and what your paper can provide to policymakers, practitioners, and researchers.

3. Please proofread the citations for the entire paper. For example, citation number 1 should be before the decimal point, not after.

4. Please add citation to lines 1 and 2.

5. Some paragraphs are below 100 words; therefore, I would suggest combination.

6. Please make sure that your paper is written in the English language, not another language. For example, Lines 594 until the end of the paragraph. Please check the whole paper.

7. Please make sure are all your figures centralized

8. Please be more rigorous when writing your papers.

Please consider these suggestions when revising your article.

Best regards,

Reviewer.

Author Response

Dear Reviewer,

thanks a lot for your review and comments to improve the article. In this cover letter, we list and explain the points of revision based on your comments. Furthermore all revisions to the manuscript are marked up can be tracked in the word file.

  • Please revise your affiliation because there is some missing information. For instance, postcode and country.

Has been edited.

  • Please add to your abstract, mention the major findings of your paper and what your paper can provide to policymakers, practitioners, and researchers.

Some information on this has been added at the end of the abstract

  • Please proofread the citations for the entire paper. For example, citation number 1 should be before the decimal point, not after.

For citation we have applied the following system regarding the character sequence: If the source refers only to one sentence, the source is placed before the period. If the source refers to several following sentences, the source is placed after the dot or at the end of a paragraph. If this is not the desired method, I would appreciate advice on where to put the source if it refers to several following sentences.

  • Please add citation to lines 1 and 2.

Citation is added now.

  • Some paragraphs are below 100 words; therefore, I would suggest combination.

We have combined the short paragraphs for the most part. There are still a few paragraphs of slightly less than 100 words left, where a summary would have made little sense.

  • Please make sure that your paper is written in the English language, not another language. For example, Lines 594 until the end of the paragraph. Please check the whole paper.

Has been edited.

  • Please make sure are all your figures centralized

Has been edited.

Best regards,
Sabine Hartmann

Round 2

Reviewer 1 Report

Dear Authors,

Thank you for the answers and improvements of the manuscript. Several comments regarding the presented manuscript are follow:

I agree with a authors that the selected problem of the topic „Hospital Construction“ is actual. In the title of the manuscript can by entitle the analysed case study in the manuscript.

In the manuscript literature analysis from the selected problem of the topic with a new literature source (from 2023- 2023 year) can be presented.

Regarding presented Keywords in „Clarivate analytics“/"Web of Science" data base with a keyword „Digitalization“ is presented 6426 number of the articles (from 2023- 2023 year);  „BIM“ -2734; „Hospital Construction“ – 10637 and etc. My recommendation to the authors to find time for the review of the manuscript. Analysis of the research articles in the review part must by the priority.

Reviewer

Author Response

Dear Reviewer,

thanks a lot for your further comments to improve our article. We were able to find some more relevant literature, even from the last two years, and include it in the paper.

Best regards,
the authors.

Reviewer 2 Report

Dear authors,

Thank you for your effort in revising the paper. Therefore, I would encourage for publication.

Warm regards,
Reviewer 

Author Response

Dear Reviewer,

thanks a lot for your review and the feedback!

Best regards,
the authors
